# Chronic Home Radon Exposure Is Associated with Higher Inflammatory Biomarker Concentrations in Children and Adolescents

**DOI:** 10.3390/ijerph20010246

**Published:** 2022-12-23

**Authors:** Brittany K. Taylor, OgheneTejiri V. Smith, Gregory E. Miller

**Affiliations:** 1Institute for Human Neuroscience, Boys Town National Research Hospital, Boys Town, NE 68010, USA; 2Department of Pharmacology and Neuroscience, Creighton University, Omaha, NE 68178, USA; 3Institute for Policy Research and Department of Psychology, Northwestern University, Evanston, IL 60208, USA

**Keywords:** CRP, IL-1β, neurotoxicity, environmental exposure, immune dysregulation, youth

## Abstract

Children are particularly vulnerable to the deleterious impacts of toxic environmental exposures, though the effects of some rather ubiquitous toxins have yet to be characterized in youths. One such toxin, radon gas, is known to accumulate to hazardous levels in homes, and has been linked with the incidence of lung cancer in aging adults. However, the degree to which chronic home radon exposure may impact risk for health problems earlier in life is unknown. Herein, we explored the degree to which chronic home radon exposure relates to biomarkers of low-grade inflammation in 68 youths ages 6- to 14 years old residing in an area of the United States prone to high home radon concentrations. Parents completed a home radon test kit, and youths provided a saliva sample to assess concentrations of five biomarkers. Using a multiple regression approach, we found that greater radon exposure was specifically associated with higher levels of C-reactive protein (β = 0.31, *p* = 0.007) and interleukin-1β (β = 0.33, *p* = 0.016). The data suggested specificity in associations between chronic home radon exposure and different biomarkers of inflammatory activity and highlight a pathway which may confer risk for future mental and physical health maladies.

## 1. Introduction

Exposures to environmental toxins can cause a host of acute- and long-term consequences on brain structure and function, which can have cascading effects on cognition, behavior, and mental wellness. This is especially true for children and adolescents [1,2]. It is well-established that youths face unique, sometimes more severe costs relative to adults following exposure to toxicants due to modes of toxicant delivery and consumption, breathing and eating habits, and biological vulnerability during periods of rapid development [1,3]. Indeed, a growing body of work has linked exposure to various toxicants, ranging from ingested heavy metals (e.g., lead) to inhaled pollutants (e.g., particulate matter), with sensory dysfunction [4], atypical patterns of structural brain development [5], incidence of specific neurodevelopmental disorders like attention-deficit/hyperactivity disorder (ADHD; [6,7]), cognitive delays and deficits [8,9], and risk for mood and anxiety disorders [10,11]. The field of research understanding the neurotoxic effects of environmental exposures on children is rapidly growing; that said, certain toxins that are known to be readily present in commonly inhabited environments have been seldom studied for their impacts on youths.

One understudied, but ubiquitous environmental toxin is radon gas. Radon is a naturally occurring, radioactive byproduct of uranium decay in soil, and is a well-established carcinogen [12,13,14]. In fact, radon inhalation is a leading cause of lung cancer globally, second only to cigarette smoking [15]. In well-ventilated environments like outdoors, as radon gas rises from the soil and enters the air, it quickly dissipates and spreads to non-injurious levels [16]. However, it is surprisingly common for radon to accumulate to hazardous concentrations over time in less well-ventilated environments like inside homes and other structures [17,18]. This issue is critically important when considering the well-established linear dose–response effect of radon exposure, whereby increased dosage and chronicity of exposure predictably relates to increasing pathophysiological consequences [13,19]. 

Across the United States (US), 1 in every 15 homes is expected to have indoor radon concentrations exceeding the action limit for mitigation defined by the US Environmental Protection Agency; this limit, established in the 1980s at or above 4 pCi/L, is the carcinogenic equivalent of smoking 10 cigarettes per day [12,16]. That said, certain areas of the country where uranium deposits in soil are greater are expected to have even further elevated risks of high radon concentrations. For instance, across eastern Nebraska and through the entire state of Iowa, at least 50% of all homes are expected to test above the action limit for indoor radon concentrations [20]. Despite the known pathophysiological risks, and the well-established pattern of high radon concentrations in many parts of the country, much of the general public is unaware of the potential consequences of radon [15,21,22]. Thus, many domiciles remain untested and unmitigated for radon, leaving dwellers chronically exposed to potentially high doses of this ubiquitous environmental toxin.

Studies in rodent models and in adult humans have repeatedly shown that alpha radiation enters the respiratory tract when radon gas is inhaled, inducing mutagenic cellular changes to mucosal tissues lining the tract and subsequently promoting upregulation of proinflammatory activities [23,24]. Studies exploring the patterns of immune dysregulation associated with radon inhalation are varied and minimal to date, though researchers have reported upregulation of an array of inflammatory proteins including several interleukins [25,26,27], and tumor necrosis factor-alpha (TNF-α; [28]). Chronic increases in circulating proinflammatory biomarkers have been associated with cascading effects on the brain, cognition, behavior, and mental and physical illness [29,30,31,32,33]. These effects can be particularly detrimental among children and adolescents, for whom the brain is highly plastic and vulnerable to physiological stress signals [34,35,36]. These associations are well-established in many contexts, but research has yet to explore the degree to which exposure to radon gas may be associated with inflammatory activity in youths and potentially increasing their risk for downstream deficits in mental and physical wellness.

The present study explored the extent to which chronic home radon exposure was associated with alterations in inflammatory activity among children and adolescents. Herein, we focus on a sample of typically developing youths residing in eastern Nebraska and western Iowa, where radon concentrations are expected to exceed the US EPA action limit for mitigation in at least 50% of dwellings [16,20]. Families completed a home radon test kit to measure individual-level home radon concentrations, and they provided information about the dwelling itself and how long the family has lived in the tested residence. Youths also provided saliva samples from which we measured concentrations of five different inflammatory biomarkers. Because of our unique interest in potential impacts on the brain, we focused on biomarkers of inflammation that have been previously implicated in structural and functional neural outcomes. We also controlled for a number of anthropometric, maturational, and socioeconomic factors that are known to influence inflammatory activity. We anticipated that, above and beyond potential confounding factors, increasing radon exposure, defined as the combination of home radon concentrations and duration of exposure, would be associated with higher levels of in inflammatory biomarkers in youths. We did not have hypotheses about specific markers that may be more strongly linked with radon given the limited literature.

## 2. Materials and Methods

### 2.1. Participants

Children and adolescents who were part of an ongoing observational study of neurocognitive development were invited to participate in the current protocol. A total of 68 youths ages 6 to 14 years-old (*M* = 10.27 years, *SD* = 2.59; 33 male) consented to the study, and their families completed a home radon test kit. Youths included in this investigation did not differ from those in the larger study on the basis of age (*t* = −1.93, *p* = 0.06), SES (*t* = −0.96, *p* = 0.34), sex (χ^2^ = 1.01, *p* = 0.32), race (χ^2^ = 4.60, *p* = 0.20), or ethnicity (χ^2^ = 3.03, *p* = 0.22). Exclusion criteria included history of head trauma, neurological disorder or other medical illness affecting brain function, current substance abuse, and standard neuroimaging exclusions (e.g., dental braces, other non-removable ferromagnetic materials on the body). All parents of child participants provided signed informed consent, and youth participants gave signed assent to participate in the study. All procedures were approved by the local Institutional Review Board.

### 2.2. Home Radon Testing

Families were provided with a commercial short-term home radon testing kit (https://www.radon.com/ , Accessed on 19 October 2022). The test kit is a standard carbon-based envelope that hangs on an interior wall on the lowest livable level of the home for three to seven days. After the testing period, the envelope is sealed and dropped in the mail for processing at the commercial lab. Parents were given the test kit along with instructions from the commercial vendor for proper exposure. We instructed families to leave the kit exposed for approximately four days. Our lab and the family each received a copy of the home radon results. In the case that a result exceeded the EPA action limit for mitigation (≥4 pCi/L), the principal investigator called the family to ensure they understood the results and provided additional information on radon safety and local resources. 

Given prior work suggesting that radon concentrations significantly vary seasonally, we provided a subset of 21 families with two radon kits. One kit was completed during summer months (June through September), and one during winter months (December through March). We compared the radon concentrations yielded from the two measurement periods to determine whether there was significant variability using a Wilcoxon Z test given the non-normal distribution of radon concentrations.

### 2.3. Questionnaires and Biometrics

Youths assessed for height and weight during a regularly scheduled lab visit. Height in centimeters and weight in kilograms were recorded for each child and used to compute each individual’s body mass index (BMI) in accordance with standard procedures.

To assess pubertal development, youths or parents were asked to complete the Pubertal Development Scale (PDS; [37]). Specifically, if the child was under the age of 11 years old, we asked the parent to complete the survey. Youths who were 11 years old and older completed the survey themselves. Surveys were completed in a private room on a computer. A trained research assistant was available to answer any questions. We computed a pubertal development stage from the PDS in accordance with recommendations by Shirtcliff, Dahl, and Pollak [38], the end result being a score parallel to Tanner staging.

Parents were asked to complete a brief questionnaire when they began their home radon testing for the study. The custom-designed questionnaire asked for details about the construction of the home (e.g., how many stories, type of foundation, location of children’s bedrooms), how long the family lived in the home, whether the home had ever been tested/mitigated for radon, and whether anyone in the home smoked cigarettes. The questionnaire was completed remotely at the parent’s convenience via the Collaborative Informatics and Neuroimaging Suite (https://coins.trendscenter.org/ , Accessed on 19 October 2022). 

In addition to the custom radon questionnaire, parents completed the Barratt Simplified Measure of Social Status [39], which assesses parental education and occupation to provide a numeric index of socioeconomic status. Scores can range from 17 to 66, with higher scores indicating higher SES. This survey was completed during a visit to the laboratory. 

### 2.4. Computing a Radon Exposure Index

Because the effects of radon exposure are cumulative, we computed a radon exposure index per participant. The index was defined as the child’s home radon concentration (in pCi/L) obtained from the home testing kit multiplied by the amount of time they lived in that home (in years). That value was natural log transformed (see Equation (1)), providing us with a normally distributed index of chronic radon exposure in their current home. Of note, we achieved similar results when computing the radon index with exposure time computed in months versus years.
Radon Exposure Index = ln ([radon concentration × exposure time] + 1)(1)

### 2.5. Saliva Sample Acquisition and Analysis

During a visit to the laboratory, youths were asked to provide a saliva sample. As such, children were told to refrain from eating, drinking, or chewing gum for at least an hour prior to sample collection. Participants were instructed to passively drool into an Oragene DISCOVER (OGR-500; www.dnagenotek.com, Accessed on 19 October 2022) until liquid (not bubble) saliva reached the indicated fill line on the tube. Thus, we collected at least 2 mL of whole unstimulated saliva from each child. Samples were stored at −20 °C until processing at the University of Nebraska Lincoln Salivary Biosciences Laboratory (https://cb3.unl.edu/sbl/ , Accessed on 19 October 2022). Samples were processed for concentrations of CRP, interleukin (IL)-1β, IL-6, IL-8, and TNF-α using commercially available assay kits (Salimetrics; www.salimetrics.com). We specifically used Meso Scale Discovery electrochemiluminescence cytokine assay kits for the interleukins and TNF-α, which provided additional sensitivity for cytokines that are naturally low in healthy populations. All assays were completed in duplicate, and the average of the two measures was used for analyses. Sensitivity, analytic range, and inter- and intra-assay coefficients of variability for each assay are listed in Table 1. In the case that a result was below the functional sensitivity for the assay, we replaced the value with half the lower limit of quantification (CRP: *n* = 7, IL-6: *n* = 1, and TNF-α: *n* = 1). The final values were then natural log transformed (ln [value + 1]) and inspected for normality. Log-transformed values exceeding three standard deviations above the group mean were excluded as outliers (CRP: *n* = 1, IL-8: *n* = 1).

### 2.6. Statistical Analysis

The main goal of the current study was to quantify the degree to which home radon exposure is associated with inflammation in children and adolescents. To address this aim, we constructed a multiple regression type model in which the radon exposure index was modeled as a predictor of each of the inflammatory markers (CRP, IL-1β, IL-6, IL-8, and TNF-α). All inflammatory markers were allowed to freely correlate. Statistical significance of hypothesized associations (i.e., those between radon exposure and each inflammatory biomarker) was determined using an α < 0.05 threshold after correction for multiple comparisons via false discovery rate (FDR). We included age, sex, BMI, and SES as control variables on each of the inflammatory measures of interest. Of note, we did attempt an alternative model in which all control variables were enforced on the radon exposure index as well, but there were no significant effects of any control variables on radon exposure. Thus, we report the model without control variables imposed on radon exposure in favor of the more parsimonious model. The conceptual model is shown in Figure 1. Several inflammatory markers were missing across participants for various reasons including outlier data, contaminated saliva samples (i.e., food residue present in sample), or inadequate saliva sample volume to complete all assays. Thus, the model was tested with and without missing data estimation using full-information maximum likelihood (FIML). Because the conclusions were the same regardless, we report the results from the analysis using FIML for robustness. As an exploratory follow-up to compare parameter estimates, we computed Z scores based on the unstandardized beta and standard error of the associations of interest (Z = [b_1_–b_2_]/√[SE_1_^2^ + SE_2_^2^]). All parameters were freely estimated using Mplus version 8.1. An a priori power analysis for a multiple regression with 6 predictors, power = 0.80, α = 0.05, and small effect size of f^2^ = 0.15 suggested a minimum sample size of 55 participants, thus we were sufficiently powered to conduct our analyses and detect relatively small effects.

## 3. Results

### 3.1. Sample Characteristics

Of the 68 youths who consented to the study, 10 did not successfully complete the home radon testing. Two of the tests were exposed for too long (i.e., longer than seven days), and the other eight took more than 11 days after completing the test exposure to arrive at the lab for processing. As such, the results from these test kits were unreliable. Additionally, one child who completed the radon testing did not provide a saliva sample. Thus, the final sample was comprised of 57 youths between the ages of 6 and 14 years-old (*M*_age_ = 10.57 years, *SD* = 2.55; 28 male). Youths’ BMIs were generally healthy, though there was notable variability as would be expected in the study sample age range (*M* = 19.04, *SD* = 5.07). SES as indexed by the BSMSS suggested that the sample was largely comprised of middle- to upper-class families, with scores ranging from 31.67 to 64.50 (*M* = 47.82, *SD* = 7.26). Of note, one family did report that one person in the home smoked cigarettes.

### 3.2. Radon and Inflammation Descriptives

Home radon test kits were exposed in participants’ homes for an average of 113.95 h (*SD* = 26.25). As expected, over half of homes tested in the current study (67%) had radon concentrations above the EPA action limit for mitigation (Figure 1). Radon levels ranged from <0.3 to 33.3 pCi/L (*M* = 6.60 pCi/L, *SD* = 7.43). As mentioned previously, a subset of 21 families completed two radon test kits: one during summer months, and one during winter months. Two of the families did take action to mitigate radon between test periods. However, even including these families in the analysis, we did not detect any significant difference in home radon concentrations acquired during summer (mean rank = 11.05, *M* = 8.53 pCi/L, *SD* = 8.10, range = 2.0–32.1 pCi/L) versus winter (mean rank = 9.95, *M* = 6.14 pCi/L, *SD* = 5.59, range = 0.3–18.0 pCi/L), Wilcoxon Z = −0.21, *p* = .84.

Youths in the study lived in their homes for 3.62 years on average (*SD* = 3.27). We used these data to compute our radon exposure index (ln ([radon concentration × exposure time] + 1)). The final index capturing chronic home radon exposure was normally distributed (*M* = 2.37, *SD* = 1.29; Figure 2). Finally, descriptive statistics for each of the inflammatory markers are reported in Table 2. Note that we report data prior to, as well as after natural log transformation.

### 3.3. Associations between Radon and Inflammation

We tested the degree to which the radon exposure index, a measure of chronic home radon exposure, was associated with each of the inflammatory markers while controlling for the potentially confounding effects of age, sex, and SES. A complete correlation matrix of the variables included in the final model is provided in Table 3. Above and beyond the effects of key demographic factors, radon exposure was significantly associated with both CRP (β = 0.31, b = 0.28, *p* = 0.007, *p*_FDR_ = 0.035) and IL-1β (β = 0.33, b = 0.19, *p* = 0.016, *p*_FDR_ = 0.040; Figure 3 and Table 4). The data suggested that youths with greater radon exposure tended to have higher concentrations of these two inflammatory markers (Figure 3). There was a small, but non-statistically significant association between radon exposure and TNF-α (β = 0.22, b = 0.11, *p* = 0.104, *p*_FDR_ = 0.173), and associations with both IL-8 and IL-6 were near zero. Finally, follow-up comparison of the parameter estimates showed that the relationship between radon exposure and CRP was significantly stronger than the association between radon exposure and IL-8 (Z = 2.10, *p* = 0.035), and was marginally stronger than the association between radon exposure and IL-6 (Z = 1.81, *p* = 0.070). 

## 4. Discussion

The present investigation explored the association between chronic radon exposure and inflammatory activity in a sample of typically developing youths residing in a region of the US that is expected to have high indoor radon concentrations. In line with EPA estimates, over half of households tested in the present study did have radon concentrations exceeding the action limit at or above 4 pCi/L. Further, our key finding was that greater radon exposure, defined as the combination of home radon concentration and chronicity of exposure, was associated with higher concentrations of both CRP and IL-1β. We also detected a similar, trending association between radon exposure and TNF-α. We discuss the implications below.

Overall, the data at least partially supported our primary hypothesis that chronic radon exposure would be associated with higher inflammatory activity in children and adolescents. This was well-aligned with prior works showing that inhaled environmental toxicants (e.g., ultrafine particulate matter) tend to stimulate inflammatory activities [40,41]. Interestingly, we demonstrated specificity in the inflammatory markers with only CRP and IL-1β significantly increasing as a function of increasing home radon exposure. There was also a trending association with TNF-α, which may be significant with a larger study sample. Importantly, these associations were detected above and beyond the effects of socioeconomic, anthropometric, and maturational effects that are known to covary with levels of circulating inflammatory activity [42,43,44]. These findings generate many new hypotheses as they suggest a potential role of home radon exposure on oral inflammation in youths, which could have implications for neural development, psychological well-being, and overall physical health.

Both CRP and IL-1β have been implicated in the pathophysiology of psychological conditions, particularly major depressive disorders [45,46,47]. For instance, a longitudinal study in adults showed that individuals with greater levels of circulating CRP at the start of the study had more severe depressive symptomology, and were more likely to have experienced a major depressive relapse nine years later [47]. In another study using rodent models, Fourrier and colleagues [48] demonstrated that overexpression of IL-1β and TNF-α was associated with disrupted learning and memory, both of which play a major role in psychiatric health. The authors suggested that the effects may be explained by alterations to neural tissues, with demonstrated reductions in synaptic plasticity in the hippocampus [48]. Numerous studies have explored the links between neuroinflammatory activities and structural brain health. Indeed, research has shown that higher levels of both CRP and IL-1β are associated with atherosclerosis in the brain, reduced white matter fractional anisotropy (i.e., microstructural integrity), and impaired synaptic plasticity and neurogenesis [48,49,50,51,52]. Further, studies have also shown altered functional neural activity coupled with inflammatory activity in circulation [53,54,55], which is also associated with decrements in cognition and psychological wellness. Of note, all of these studies have focused on systemic inflammatory activity, and it remains unclear whether the patterns we observed here—in oral tissue—would have similar consequences for the brain.

Recent reviews do suggest that chronically elevated inflammatory activity early in life can and often does have long-term consequences [35,56]. This is critical when considering a new and burgeoning line of research linking chronic radon exposure to neurodegenerative diseases later in life, including multiple sclerosis, Alzheimer’s Disease and related dementias, and Parkinson’s Disease [57,58,59,60]. It is possible that chronic radon exposure starting early in life may, at least in part, contribute to later life neuropathologies due to continuously elevated inflammation. Further work is needed to understand if and how neuroinflammatory profiles may mediate long-term effects of radon exposure on neurological health.

With respect to physical health, inflammation induced by radon exposure, among other potential pollutants, is known to cause cascading effects on immune health overall and can confer risk for physical illnesses [61,62]. For instance, given the damage caused to respiratory epithelium, inhaled radon gas has been linked to the incidence of numerous respiratory diseases in childhood well before any type of carcinogenesis is detected; among the most common health maladies are asthma and chronic obstructive pulmonary disease (COPD) [62,63]. Such illnesses are associated with increasing lifetime burdens for the individual and for society, including school- and work-related losses, medical costs, limitation of physical activities, and overall lower quality of life [64]. Importantly, these chronic conditions are known to be exacerbated in the context of combined exposures to radon and other pollutants (e.g., phthalates, secondhand smoke; [65]). Given that radon is easily mitigated from homes, this seems an easy target to potentially quell the impacts of these lifelong conditions and improve quality of life for a large sect of the population. 

In contrast to prior work [25,26,27], we did not detect associations between radon exposure and either IL-6 or IL-8. Typically, IL-6 has been associated with radon exposure under extreme circumstances. For example, Leng et al. [25] explored IL-6 as a promotor of lung squamous cell carcinoma in former uranium miners who would have been chronically exposed to extreme levels of radon and other potentially toxic compounds in their working environment. Similarly, Narayanan and colleagues [27] studies IL-8 in vitro in alpha-irradiated lung fibroblasts. Such conditions are not necessarily representative of effects in living humans, nor do they reflect the more normative, domestic radon exposure effects that we studied herein. Even so, future works should still consider IL-6 and IL-8 in the context of radon exposure given their known responses to other forms of ionizing radiation [66]. It is possible that the model we tested in our study did not adequately capture the roles of these interleukins. Van Der Meeren and colleagues [67] suggested that IL-6 and IL-8 might be better described as inflammatory mediators, serving as an important process linking environmental exposures and other downstream proinflammatory activities. More recent works have also demonstrated unique and interactive effects of different isoforms of CRP with both IL-6 and IL-8 [68]. Future works would require larger samples to adequately model the potentially complex mediating and moderating effects of different inflammatory activities following chronic home radon exposure.

Another contrasting body of work using controlled dosing in rodent models suggests that radon inhalation may actually inhibit proinflammatory activities by increasing the body’s production of reactive oxygen species (ROS; e.g., [69,70]). Indeed, some work suggests potential therapeutic effects of brief, low-level radon exposures in specialized radon spas, particularly for relieving symptoms of inflammatory conditions like rheumatoid arthritis [71,72]. From this and dissenting literature, it is clear that the effects of radon on the body are complex in nature and may vary based on dosage, frequency, and mode of exposure (i.e., inhaled, ingested, etc). Further work is needed to better characterize the full effects of radon exposure on the human immune system. 

Generally speaking, the literature provides sound evidence for the biological plausibility that cytokines may circulate in higher quantities in the presence of persistent environmental stressors (e.g., [73,74], including exposure to inhaled toxicants like radon (e.g., [25,27,28]). That said, increases in circulating cytokines is likely not the complete story. Biochemically, many cytokines, including those measured in the current investigation, have relatively short half-lives ranging from 18 min to roughly 30 h depending on the model organism and measurement modality [75,76]. Although the continued exposure to radon gas would likely perpetuate the production of these inflammatory biomarkers, their effects downstream on other biochemical processes like oxidative stress and damage to DNA [60,70,77,78] likely also play important roles in any noted effects of radon on the functioning system/ Future investigations should incorporate additional measurements of the biological impacts of radon exposure in vivo in children.

Before closing, it is important to recognize the limitations of the study. First off, radon was only assessed through short-term home radon test kits (three- to seven-day measurement period). Although the kits utilized herein are generally reliable, they are susceptible to inaccuracies due to intermediary factors including adverse weather at the time of testing, open doors or windows during the exposure period, and damp conditions in the location of the test. Further, historical ecidence suggests that radon concentrations vary across seasons, which would not be captured with such short-term measurements. Although we did not detect seasonal variability in the subset of participants who completed tests in both summer and winter months, this was a small sample and requires additional verification. Deploying multiple test kits simultaneously during multiple seasons would help solidify the reliability of measurements obtained, and help clarify the degree of seasonal variability in the local area; this is a study design limitation that could be easily addressed in future works. Alternatively, studies do suggest that longer-term kits exposed anywhere from 30 days to one year can more accurately measure radon concentrations within a dwelling [21], and these tests do tend to be quite robust against the aforementioned conditions. Second, the test kits used herein only assessed current home radon concentrations. Although we did approximate cumulative exposure based on the current home radon concentration and the amount of time the child had lived in the home, there may be better ways to assess lifetime exposure more accurately, especially if the child has lived in multiple homes for which we had no radon data. It is possible with specialized equipment to measure historic home radon concentrations if the family is able to provide a sample of glass that has been in the dwelling for the desired timeframe (i.e., the child’s lifetime; [79,80,81]). Another limitation is that we only measured radon concentrations in homes. Although the home is the greatest source of indoor radon exposure for most people [82], children are exposed to radon in other venues including schools. Future studies may consider adding measurements of other commonly occupied facilities as well. 

Aside from the radon measurements, there were several other limitations to this study. The investigation was cross-sectional, but a longitudinal design would allow for clearer linkages between the chronicity of radon exposure and inflammatory activity. Additionally, inflammatory markers were only assessed using saliva samples, which can be contaminated by oral health, and whose relevance for brain development and psychiatric illness is less clear. Future investigations can address these issues by measuring inflammatory biomarkers in circulating blood, or in biopsied tissue procured medical procedures (e.g., bronchoscopy). Finally, we did not have information on potential physical comorbidities that could have impacted inflammation and/or the effects of radon (e.g., asthma), and we did not control for other pollutants like secondhand smoke exposure (note that only one family reported any smokers in the home). Future investigations would benefit from additional controls for such comorbidities and concomitant exposure.

## 5. Conclusions

To conclude, the findings of the present study contribute to a growing body of work exploring the inflammatory consequences of an array of environmental toxic exposures. The work we presented herein focused on the inflammatory effects of a specific toxin, home radon, which has been sorely understudied for its potential effects on developing youths. We found significantly elevated concentrations of multiple inflammatory biomarkers as a function of increasing home radon exposure. These data provide insight into the biological effects of chronic radon exposure and support the need for further characterization of the effects of this ubiquitous environmental toxin on the developing body. 

## Figures and Tables

**Figure 1 ijerph-20-00246-f001:**
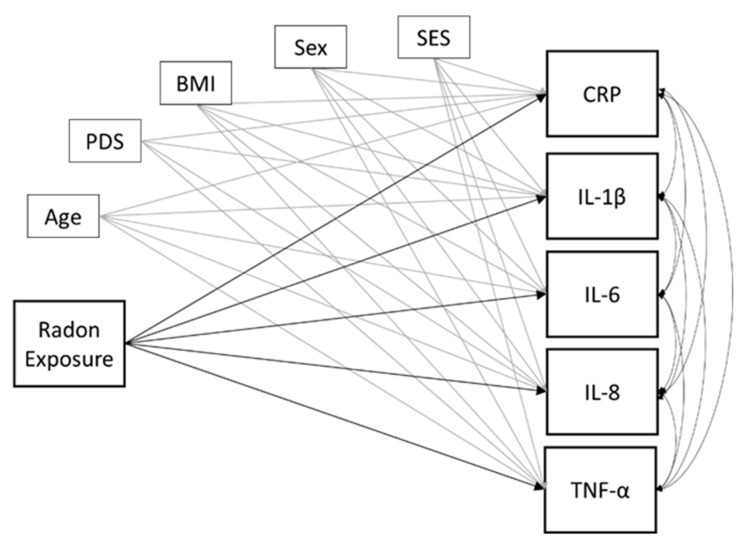
Conceptual model. Radon exposure is modeled as a predictor of each of the inflammatory markers of interest. Age, sex, SES, pubertal development, and BMI serve as control variables on all inflammatory markers. Single-headed arrows indicate directional, predictive relationships. Double-headed curved arrows show correlations. For clarity, control associations are shown in gray, and predictive relationships of interest are shown in black. Sex was dummy coded as 0 = “male”, 1 = “female”. “BMI” = body mass index; “CRP” = c-reactive protein; “IL” = interleukin; “PDS” = pubertal development stage; “SES” = socioeconomic status; “TNF-α” = tumor necrosis factor alpha.

**Figure 2 ijerph-20-00246-f002:**
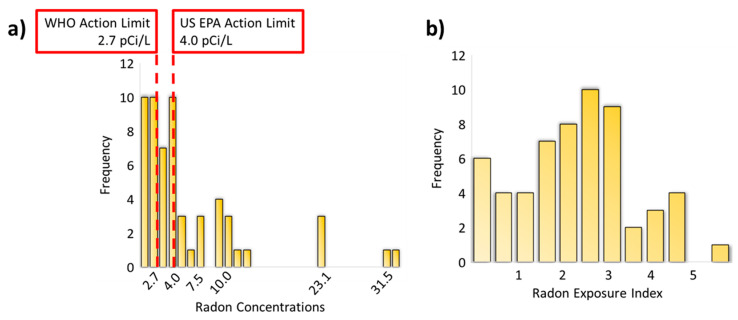
Histogram of acquired home radon concentrations (**a**) and of the computed radon exposure index (**b**). The left histogram is marked with the WHO and the US EPA action limits for indoor radon concentrations (at or above 2.7 and 4.0 pCi/L, respectively). The radon exposure index, calculated at the natural log of home radon concentrations multiplied by the amount of time the person has lived in their home (+1), is relatively normally distributed.

**Figure 3 ijerph-20-00246-f003:**
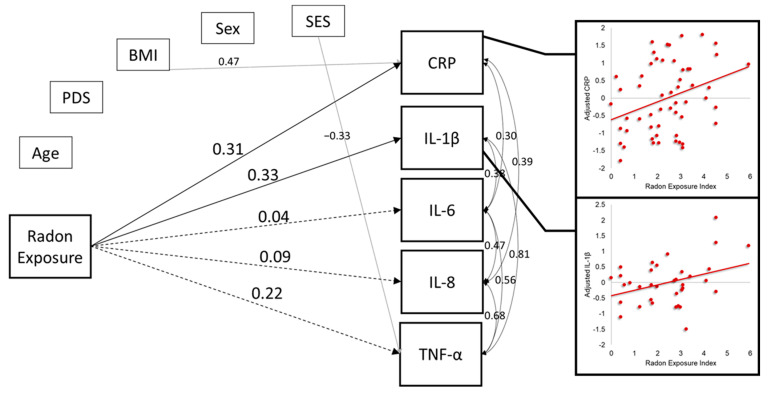
Model results. For simplicity, only associations of interest between radon exposure and inflammatory markers (regardless of statistical significance), and other effects significant at the *p*_FDR_ < 0.05 level are shown. All inflammatory markers were natural log transformed. All parameters reported in the figure are standardized coefficients. Solid lines indicate statistically significant associations at the *p*_FDR_ < 0.05 level, and dashed lines show non-statistically significant associations (*p*_FDR_ > 0.05). Scatterplots show the statistically significant associations between the radon exposure index and natural log transformed CRP (top), and IL-1β (bottom) after adjusting for the effects of age, sex, pubertal development, BMI, and SES. Sex was dummy coded as 0 = “male”, 1 = “female”. “BMI” = body mass index; “CRP” = c-reactive protein; “IL” = interleukin; “PDS” = pubertal development stage; “SES” = socioeconomic status; “TNF-α” = tumor necrosis factor alpha. Scatterplots of significant associations between radon exposure and inflammatory markers.

**Table 1 ijerph-20-00246-t001:** Characteristics of assays for inflammatory markers in the current study.

	Assay Range	Analytical Sensitivity	Functional Sensitivity	Intra-Assay CV	Inter-Assay CV
CRP	25–1600 pg/mL	0.042 pg/mL	19.44 pg/mL	1.93%	3.58%
IL-1β	0.646–375 pg/mL	0.05 pg/mL	0.646 pg/mL	2.65%	4.98%
IL-6	0.633–488 pg/mL	0.06 pg/mL	0.633 pg/mL	3.85%	2.88%
IL-8	0.591–375 pg/mL	0.07 pg/mL	0.591 pg/mL	1.31%	3.12%
TNF-α	0.690–248 pg/mL	0.04 pg/mL	0.690 pg/mL	4.90%	3.19%

**Table 2 ijerph-20-00246-t002:** Descriptive statistics for each of the inflammatory markers (pg/mL) before and after natural log transformation.

	Original (pg/mL)			Natural Log Transformed		
	*M*	*SD*	*Range*	*M*	*SD*	*Range*
CRP	319.74	1651.54	9.72–12,466.16	3.96	1.36	2.37–9.43
IL-1β	115.00	134.32	24.03–791.56	4.43	0.75	3.22–6.68
IL-6	7.21	10.23	0.32–58.79	1.67	0.86	0.27–4.09
IL-8	881.89	675.63	280.02–4292.42	6.62	0.54	5.64–8.36
TNF-α	5.10	5.18	0.35–25.76	1.58	0.65	0.30–3.29

**Table 3 ijerph-20-00246-t003:** Correlations among all variables included in the model.

	1	2	3	4	5	6	7	8	9	10	11
**1. Radon Exposure**	–										
**2. CRP**	0.34	–									
**3. IL-1β**	0.28	0.32	–								
**4. IL-6**	0.10	0.20	0.40	–							
**5. IL-8**	0.12	0.33	0.53	0.45	–						
**6. TNF-α**	0.28	0.21	0.79	0.58	0.68	–					
**7. Age**	0.20	0.21	0.13	0.10	−0.08	−0.04	–				
**8. PDS**	0.13	0.36	0.17	−0.02	−0.05	−0.12	0.72	–			
**9. Sex**	−0.10	0.10	−0.03	−0.07	−0.27	−0.29	0.08	0.40	–		
**10. SES**	−0.18	−0.01	−0.28	−0.29	−0.08	−0.34	−0.14	−0.05	−0.01	–	
**11. BMI**	0.12	0.51	0.17	−0.04	−0.01	−0.06	0.46	0.64	0.25	−0.11	–

Note: All inflammatory markers were natural log transformed; Sex was dummy coded as 0 = “male”, 1 = “female”. “BMI” = body mass index; “CRP” = c-reactive protein; “IL” = interleukin; “PDS” = pubertal development stage; “SES” = socioeconomic status; “TNF-α” = tumor necrosis factor alpha.

**Table 4 ijerph-20-00246-t004:** Complete model results exploring associations between radon exposure and inflammatory markers, controlling for age, sex, SES, BMI, and PDS.

Path	β	b	SE	b/SE	*p*
**CRP On**					
**Radon Exposure**	**0.307**	**0.277**	**0.103**	**2.682**	**0.007**
Age	−0.176	−0.080	0.076	−1.053	0.292
Sex	−0.040	−0.093	0.290	−0.322	0.747
SES	0.076	0.012	0.018	0.686	0.492
PDS	0.169	0.160	0.193	0.827	0.408
**BMI**	**0.466**	**0.106**	**0.032**	**3.264**	**0.001**
**IL-β on**					
**Radon Exposure**	**0.328**	**0.191**	**0.079**	**2.405**	**0.016**
Age	−0.158	−0.046	0.079	−0.583	0.560
Sex	−0.102	−0.152	0.297	−0.512	0.609
SES	−0.225	−0.023	0.016	−1.455	0.146
PDS	0.246	0.150	0.192	0.783	0.434
BMI	0.047	0.007	0.025	0.274	0.784
**IL-6 on**					
Radon Exposure	0.044	0.028	0.090	0.322	0.747
Age	0.133	0.045	0.066	0.675	0.499
Sex	−0.035	−0.061	0.262	−0.231	0.817
**SES**	**−0.279**	**−0.033**	**0.016**	**−2.121**	**0.034**
PDS	−0.061	−0.043	0.168	−0.254	0.799
BMI	−0.086	−0.014	0.028	−0.515	0.606
**IL-8 on**					
Radon Exposure	0.087	0.030	0.057	0.532	0.595
Age	−0.287	−0.050	0.048	−1.044	0.297
Sex	−3.49	−0.313	0.186	−1.686	0.092
SES	−0.092	−0.006	0.010	−0.565	0.572
PDS	0.272	0.100	0.119	0.837	0.403
BMI	0.011	0.001	0.019	0.054	0.957
**TNF-α on**					
Radon Exposure	0.217	0.105	0.064	1.626	0.104
Age	−0.146	−0.036	0.064	−0.559	0.576
Sex	−0.284	−0.352	0.230	−1.535	0.125
**SES**	**−0.326**	**−0.028**	**0.012**	**−2.241**	**0.025**
PDS	0.073	0.037	0.154	0.240	0.811
BMI	−0.029	−0.003	0.020	−0.173	0.863
**CRP with**					
IL-1β	0.186	0.114	0.103	1.107	0.268
**IL-6**	**0.299**	**0.225**	**0.107**	**2.095**	**0.036**
**IL-8**	**0.390**	**0.152**	**0.068**	**2.232**	**0.026**
TNF-α	0.256	0.127	0.082	1.550	0.121
**IL-1β with**					
**IL-6**	**0.376**	**0.201**	**0.095**	**2.111**	**0.035**
**IL-8**	**0.539**	**0.149**	**0.054**	**2.747**	**0.006**
**TNF-α**	**0.807**	**0.284**	**0.077**	**3.668**	**<0.001**
**IL-6 with**					
**IL-8**	**0.471**	**0.161**	**0.063**	**2.539**	**0.011**
**TNF-α**	**0.561**	**0.243**	**0.079**	**3.091**	**0.002**
**IL-8 with**					
**TNF-α**	**0.675**	**0.152**	**0.048**	**3.190**	**0.001**

Note: Sex was dummy coded as 0 = “male”, 1 = “female”; “BMI” = body mass index; “CRP” = c-reactive protein; “IL” = interleukin; “PDS” = pubertal development stage; “SES” = socioeconomic status; “TNF-α” = tumor necrosis factor alpha all inflammatory markers were natural log transformed; statistically significant effects at the *p* < 0.05 level are bolded for clarity.

## Data Availability

The data used herein will be made publicly available on request via the Collaborative Informatics and Neuroimaging Suite (COINS; www.coins.trendscenter.org, Accessed on 19 October 2022) after conclusion of the study data collection.

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
