# Peer review of "Chronic Home Radon Exposure Is Associated with Higher Inflammatory Biomarker Concentrations in Children and Adolescents"

_ijerph, 2022, doi:10.3390/ijerph20010246_

Round 1
Reviewer 1 Report
The authors examine the correlation between short-term radon tests and salivary levels of several cytokines. They report that, in 58 children, salivary markers of 2 of the 5 cytokines were significantly correlated and conclude that long term radon exposure is a risk for elevated cytokines.
The paper is interesting in concept, but has several weaknesses that detract from its ability to persuade readers of the validity of the conclusions. It also has several problems with regard to presentation. In order of appearance, some of these issues are:
The Abstract, in Line 16, asserts that radon has been linked to neurodegenerative diseases. This is presented without evidence or citation. I do not believe that there is any convincing evidence of this (and an ecologic study or hypothesis study, or which there are numerous, do not constitute evidence for an effect in individuals). Similarly, there is no evidence for mental health effects, as noted in line 25. These assertions should either be supported by evidence or deleted.
The Introduction is overly long and should be shortened by about 50%. Material that is only tangentially related to the actual data presented, such as the discussion on cancer, beginning on line 68, is not really relevant and the authors can cut straight to the issue of inflammation, beginning on line 80.
Methods. The authors use a single short-term radon test, yet claim that they are measuring chronic radon exposure. This claim needs some support. For example, it is well known that, esp in the regions of the US where the authors measured radon, that radon levels are significantly higher in winter than in other times of the year. Unless I missed it, there does not seem to be any evidence provided on seasonality of radon measurement, so it this is an important potential confounder left unaddressed. Were all the homes measured at the same time? If not, were the radon levels adjusted for season? Similarly, the Radon Exposure Index is problematic. For example, if a subject lived for say, 5 years in a highly radon exposed home but then moved to a well-ventilated low radon exposed home in the last year, his/her radon exposure index would not reflect the actual history of radon exposure. This is a limitation that is not addressed in the Discussion.
More importantly, the issue of biological credibility of the claimed associations is not addressed. As noted above, the authors are claiming that a short-term radon test reflects long-term radon exposure. That is itself debatable. However, it not to clear to me how the measure of a cytokine such as CRP, whose half-life in serum is 19 hours, would be related to long term radon levels. This needs to be addressed, and in that regard, information on the half-lives of all of the cytokines studies should be provided.
Another important question that requires attention is that of Why were these particular cytokines chosen? If there indeed was a significant relationship between radon and salivary cytokine levels, does an effect for 2 of the 5 cytokines really support the authors' initial hypothesis? I believe that these results are more hypothesis generating than hypothesis testing, and I would like the authors to reflect on this explicitly.
Finally, the authors do not adequately address literature that is not consistent with the results of their observational study. For example, Kataoka T, et al., have published several studies in which animals are exposed to radon in controlled situations. Many of these studies show an inhibitory effect of radon on cytokine levels. These discrepant results should at least be acknowledged.
Author Response
Please see attachment for our response to the reviewer's comments.

Reviewer 2 Report
see the attached file.

Author Response

(The authors gave the same response as above.)

Round 2
Reviewer 1 Report
The authors do not adequately address the issue of seasonality or the limits of their attempt to control it.
Their subset of 18 or 19 families of a total > 60 has very little power to detect seasonal differences, yet they present their findings of a lack of seasonality as evidence that it does not exist. At the very least, the power of this analysis, or lack thereof, must be listed as a limitation.
There is inadequate discussion of the issue of biological plausibility. The half-lives of the cytokines listed is not discussed, despite the request to provide this.
Reviewer 2 Report
For the most part, the authors responded well to the concerns and comments. See my comments on the attached response document.
